# Clinical utility of mono-exponential model diffusion weighted imaging using two b-values compared to the bi- or stretched exponential model for the diagnosis of biliary atresia in infant liver MRI

**Jisoo Kim[1], Haesung Yoon[1,2], Mi-Jung Lee[1,2], Myung-Joon Kim[1,2], Kyunghwa Han[1], Seok Joo Han[2,3], Hong Koh[2,3], Seung Kim[2,4], Hyun Joo Shin[1,2]***

1 Department of Radiology, Severance Hospital, Research Institute of Radiological Science, Yonsei University College of Medicine, Seoul, Korea, 2 Severance Pediatric Liver Disease Research Group, Severance Hospital, Yonsei University College of Medicine, Seoul, Korea, 3 Department of Pediatric Surgery, Severance Hospital, Yonsei University College of Medicine, Seoul, Korea, 4 Department of Pediatric Gastroenterology, Hepatology and Nutrition, Severance Hospital, Yonsei University College of Medicine, Seoul, Korea

* lamer-22@yuhs.ac

## Abstract

### Purpose

To investigate the clinical utility of mono-exponential model diffusion weighted imaging (DWI) using two b-values compared to the bi- or stretched exponential model to differentiate biliary atresia (BA) from non-BA in pediatric liver magnetic resonance imaging (MRI).

### Methods

Patients who underwent liver MRI with DWI for suspected BA from November 2017 to September 2018 were retrospectively included and divided into BA and non-BA groups. Laboratory results including γ-glutamyl transferase (γGT) were compared between the two groups using the Mann-Whitney U test and Fisher's exact test. The hepatic apparent diffusion coefficient (ADC) 10 using ten b-values and ADC 2 using two b-values were obtained from the mono-exponential model. The slow diffusion coefficient (D), fast diffusion coefficient (D*), and perfusion fraction (f) were obtained from the bi-exponential model. The distributed diffusion coefficient (DDC) and heterogeneity index (α) were measured from the stretched exponential model. Parameters were compared between the two groups using a linear mixed model and diagnostic performance was assessed using the area under the curve (AUC) analysis.

### Results

For 12 patients in the BA and five patients in the non-BA group, the ADC 10 (median 0.985 ×10$^{-3}$ mm$^2$/s vs. 1.332 ×10$^{-3}$ mm$^2$/s, p = 0.008), ADC 2 (median 0.987 ×10$^{-3}$ mm$^2$/s vs.

**Data Availability Statement:** All relevant data are within the manuscript and its Supporting Information files.

**Funding:** The authors received no specific funding for this work.

**Competing interests:** The authors have declared that no competing interests exist.

$1.335 \times 10^{-3}$ mm$^2$/s, p = 0.017), D* (median $33.2 \times 10^{-3}$ mm$^2$/s vs. $55.3 \times 10^{-3}$ mm$^2$/s, p = 0.021), $f$ (median 13.4%, vs. 22.1%, p = 0.009), and DDC (median $0.889 \times 10^{-3}$ mm$^2$/s vs. $1.323 \times 10^{-3}$ mm$^2$/s, p = 0.009) values were lower and the γGT (median 368.0 IU/L vs. 93.5 IU/L, p = 0.02) and α (median 0.699 vs. 0.556, p = 0.023) values were higher in the BA group. The AUC values for γGT (AUC 0.867 95% confidence interval [CI] 0.616–0.984), ADC 10 (AUC 0.963, 95% CI 0.834–0.998), ADC 2 (AUC 0.925, 95% CI 0.781–0.987), $f$ (AUC 0.850, 95% CI 0.686–0.949), and DDC (AUC 0.925, 95% CI 0.781–0.987) were not significantly different, except for the D* and α values.

## Conclusion

Patients with BA had lower ADC 10, ADC 2, D*, $f$, and DDC values and higher γGT and α values than those in the non-BA group. The diagnostic performance of ADC 2 using only two b-values showed excellent diagnostic performance and was not significantly different from that of γGT, ADC 10, f, and DDC for diagnosing BA.

## Introduction

Biliary atresia (BA) is a rare but fatal cholestatic disease of infants caused by fibro-obliteration of the biliary tree [1]. Early diagnosis and intervention with Kasai portoenterostomy are essential to prevent the progression of liver fibrosis[2]. Delayed diagnosis or complications from intractable cholangitis and portal hypertension could necessitate liver transplantation in children [1,3]; therefore, early and accurate diagnosis of BA is important. For the diagnosis of BA, gray-scale ultrasonography (US), shear wave elastography, hepatobiliary scan, and magnetic resonance imaging (MRI) can be used in addition to the clinical and laboratory tests for neonates presenting with jaundice [2,4]. Even though there are fewer MRI studies compared to US, MRI has proven advantages over US due to its operator-independency, reproducibility and unrestricted field-of-view and it permits visualization of bile ducts and periportal fibrotic masses in BA patients [4–7].

Another advantage of MRI is its quantitative imaging functions. Diffusion weighted imaging (DWI) is a widely used technique; it measures the degree of diffusion of water molecules using a mono-exponential model (MEM) [8]. Separation of pure water molecular diffusion and microvascular perfusion from MEM became possible by adopting multiple b-values, which is now known as the bi-exponential model (BEM) or intravoxel incoherent motion (IVIM) technique [9]. Recently, the stretched exponential model (SEM) was introduced and takes into account the heterogeneous nature of diffusion in different tissues by measuring signal attenuation deviation from mono-exponential values [10].

In previous studies, the apparent diffusion coefficient (ADC) values from MEM were significantly lower in the liver of BA patients compared to patients with neonatal hepatitis [11–13]. A previous study demonstrated that ADC could be a new imaging parameter for assessing liver fibrosis in BA patients [12]. In children with nonalcoholic fatty liver disease, molecular diffusion and perfusion parameters from BEM were differently affected by hepatic steatosis and fibrosis [14]. In a recent study, SEM showed better diagnostic performance for diagnosing hepatic fibrosis in adults [15]. However, to our knowledge, there are no studies investigating the utility of BEM and SEM to diagnose BA in children. In addition, no study has compared the diagnostic performance of various DWI sequences using different models and b-values to reduce the acquisition time of MRI in young infants.

Therefore, the purpose of this study was to assess the clinical utility of mono-exponential model DWI using two b-values compared to the bi- or stretched exponential model to differentiate BA from non-BA in pediatric liver MRI.

## Material and methods

### Subjects

This retrospective study was approved by the Institutional Review Board of Severance Hospital (Protocol no. 1-2018-0076) and informed consent was waved. Children who underwent liver MRI for suspected BA due to hyperbilirubinemia from November 2017 to September 2018 were included. We excluded children who had undergone liver MRI DWI sequences with different compositions of b values (DWI MRI without using b-values of 0, 25, 50, 75, 100, 150, 200, 400, 600, and 800 s/mm$^2$) or who had other hepatic lesions on MRI.

Subjects were divided into two groups according to their operative procedure and pathologic results: BA and non-BA-groups. Diagnosis of BA or non-BA was made through intraoperative cholangiography or liver biopsy. Age in weeks, gender, and laboratory results including aspartate aminotransferase (AST, IU/L), alanine transaminase (ALT, IU/L), total bilirubin (mg/dl), direct bilirubin (mg/dl), alkaline phosphatase (ALP, IU/L), and γ-glutamyl transferase (γGT, IU/L) were assessed within 3 days from the time of MRI examination.

### MRI acquisition

Liver MRI was performed with a 1.5 T system (Achieva dStream; Philips Healthcare, Best, the Netherlands) in a pediatric body coil, all of which were performed prior to procedures such as biopsy or cholangiography. MR examinations were performed under sedation administered by a trained pediatric sedation team, and the patients were in free-breathing status. Free-breathing DWI was performed using 10 b-values (0, 25, 50, 75, 100, 150, 200, 400, 600, and 800 s/mm$^2$) with single-shot spin-echo echo-planar imaging (SE EPI) using gradient reversal fat suppression. The used MRI parameters for DWI sequence was as follows; repetition time (TR) 4000 msec, echo time (TE) 90 msec, matrix 128×128, slice thickness 3 mm, flip angle 90˚, and number of signal averages 3. The total acquisition time of DWI was 5 minutes 16 seconds.

### Diffusion parameters analysis

From the DWI sequence, ADC values can be calculated using the following equation for MEM [8,16]:

$$S/S0 = \exp(-b \times ADC) \qquad (1)$$

where S represents the degree of signal attenuation, S0 means signal intensity of the T2-weighted image with no diffusion gradient applied, and b value means degree of diffusion weighting.

Using BEM, D (slow true diffusion from pure water molecular diffusion), D* (fast pseudo-diffusion from microcirculation and perfusion), and $f$ (perfusion fraction) values were obtained using the following equation [17]:

$$S/S_0 = \{f \cdot \exp(-b \cdot D*) + \{(1-f) \cdot \exp(-b \cdot D)\}. \qquad (2)$$

For SEM, heterogeneity of diffusion was assessed by measuring the deviation of diffusion from the mono-exponential behavior using the following equation [10,15]:

$$S/S0 = \exp\{-(b \times DDC)\alpha\}. \qquad (3)$$

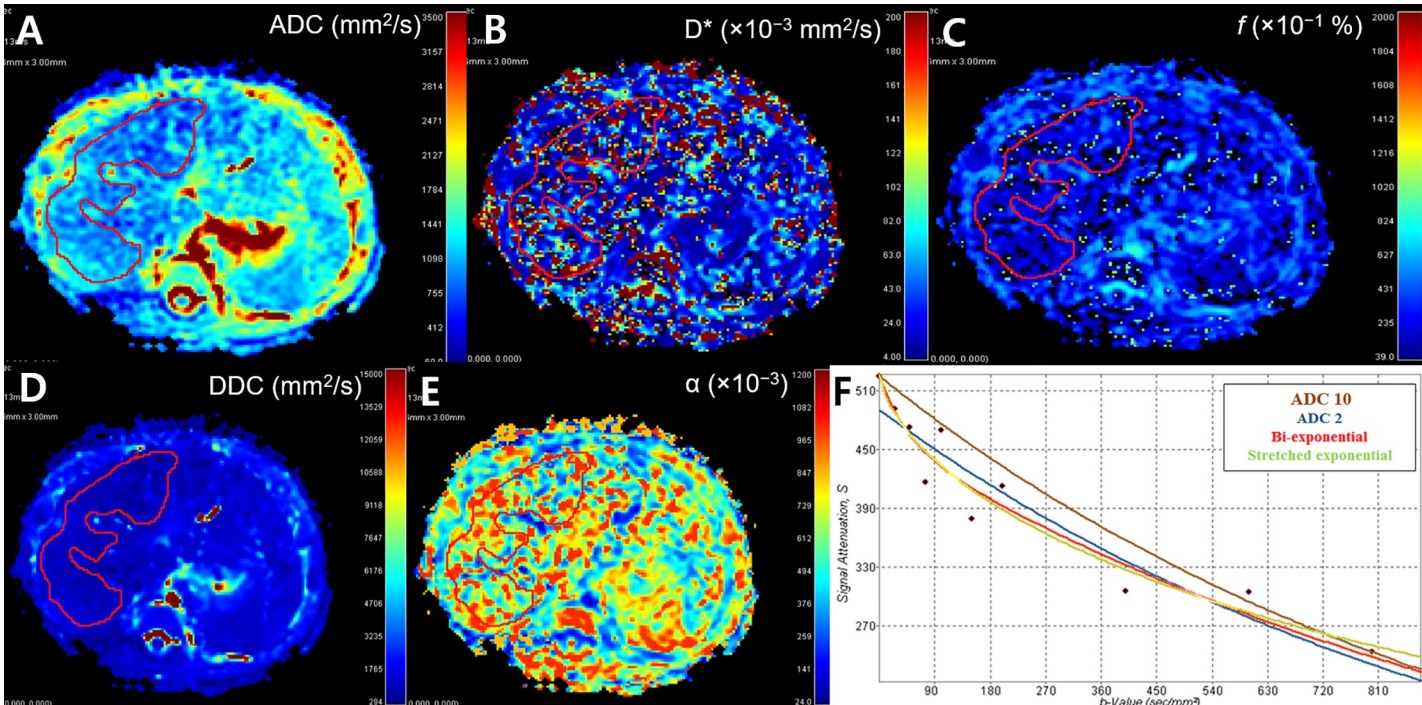

**Fig 1. Liver diffusion MRI images of a 5-week-old girl in the BA group.** Her initial total/direct bilirubin levels were 5.7/4.7 mg/dl, and her $\gamma$GT level was 403 IU/L. (A) The ADC with two b-values was $0.939 \times 10^{-3}$ mm$^2$/s. (B) The D$^*$ value was $22.6 \times 10^{-3}$ mm$^2$/s. (C) The $f$ value was 8%. (D) The DDC value was $0.839 \times 10^{-3}$ mm$^2$/s. (E) The $\alpha$ value was 0.757. (F) Multi-parametric curves from the mono-, bi-, and stretched exponential models with the x-axis representing b-values and the y-axis representing diffusion-related signal attenuation.

The distributed diffusion coefficient (DDC) means intravoxel diffusion rate in the presence of heterogeneity of diffusion without separating proton pools into the compartments. The diffusion heterogeneity index ($\alpha$) demonstrates the degree of deviation of diffusion signal intensities from mono-exponential curve and ranges from 0 to 1 [10]. When the $\alpha$ value is close to 1, the equation follows a mono-exponential curve, and it demonstrates homogeneous diffusion compositions of the proton pools in environment. When the $\alpha$ value is close to 0, it means high intravoxel diffusion heterogeneity [10].

To acquire parametric maps and parameters, we used EXPRESS software version 2.0 (Philips Healthcare, Andover, MA, USA). By loading raw data of DWI sequences in par/rec file formats in this software, multi-parametric maps including ADC 10 using all of the 10 b-values and ADC 2 using two b-values (0, 800 s/mm$^2$), BEM or IVIM with asymptotic fitting (calculated D, then fit D$^*$ and f), and SEM were automatically presented. To maintain uniformity, one pediatric radiologist (H.J.S.) who was blind to the final diagnosis selected two representative axial images at the main portal vein level of each patient, and plotted the freehand region-of-interests (ROIs) to cover as much liver parenchyma as possible on the images while avoiding major vessels (Fig 1). Parameters including ADC 10, ADC 2, D, D$^*$, $f$, DDC, and $\alpha$ values were automatically calculated from each ROI.

## Statistical analysis

Statistical analyses were performed using SPSS version 23 (IBM Corp., Armonk, NY, United States) and MedCalc version 18.2.1 (Ostend, Belgium). Clinical and laboratory results were compared between BA and non-BA groups using Mann-Whitney U and Fisher's exact tests as

appropriate. The area of ROIs (mm$^2$) and diffusion parameters were compared using a linear mixed model (LMM) because two measurements were repeated using two axial images for each patient. The BA diagnostic performances were assessed and compared using area under the curve (AUC) analysis. The optimal cutoff value of each parameter was chosen to maximize the sum of sensitivity and specificity in AUC analysis. To compare the receiver operating characteristic (ROC) curves, the DeLong method was used with the MedCalc program. In addition, the logistic regression test and AUC analysis were performed to analyze diagnostic performances combining ADC 2 and γGT. All data are presented as median values and interquartile ranges. Estimated average values with 95% confidence intervals (CIs) are presented when using LMM. P-values less than 0.05 were considered statistically significant.

## Results

### Subjects and comparison of clinical and laboratory results

During the study period, a total of 19 children suspected of BA underwent liver MRI. Among them, one child was excluded due to a DWI sequence with different composition of b-values, and one child was excluded because he had multiple liver hemangiomas. Therefore, seventeen patients (M:F = 8:9, mean age 8.8 weeks old, range of 5–16 weeks) were included in this study. Among them, 12 of 17 patients (70.6%) were assigned to the BA group, while five patients (29.4%) were included in the non-BA group because they were confirmed to have neonatal hepatitis (n = 2), total parenteral nutrition induced cholestasis (n = 1), alagille syndrome (n = 1), and Dubin-Johnson syndrome (n = 1). The clinical and laboratory results for the BA and non-BA groups are shown in Table 1. Age and gender were not significantly different between the two groups (p = 1.000, 0.131, respectively). Among laboratory results, only the γGT level was significantly higher in the BA compared to the non-BA group (median 368.0 IU/L vs. 93.5 IU/L, p = 0.020).

### Comparison of diffusion parameters

Table 2 shows the results of diffusion parameters for the BA and non-BA groups calculated from MEM, BEM, and SEM. The ROI area was not significantly different between the two groups (median 1215.5 mm$^2$ vs. 1256.2 mm$^2$, p = 0.810). From MEM, the ADC 10 (median

**Table 1. Comparison of clinical and laboratory results between BA and non-BA groups.**

|  | BA (n = 12) | non-BA (n = 5) | p-value |
|---|---|---|---|
| Age (weeks) | 8 (6, 12) | 8 (5, 11) | 1.000 |
| Gender (M:F) | 4:8 | 4:1 | 0.131' |
| AST (IU/L) | 168.5 (112.8, 374.3) | 72.5 (35.0, 148.3) | 0.114 |
| ALT (IU/L) | 101.5 (53.8, 173.5) | 27.5 (15.8, 103.0) | 0.206 |
| Total bilirubin (mg/dl) | 8.0 (5.7, 9.1) | 5.75 (3.8, 11.4) | 0.527 |
| Direct bilirubin (mg/dl) | 6.0 (4.6, 6.9) | 3.20 (3.0, 7.8) | 0.291 |
| ALP (IU/L) | 649.5 (465.8, 795.3) | 616.5 (354.5, 1006.0) | 0.752 |
| γGT (IU/L) | 368.0 (185.8, 545.0) | 93.5 (86.5, 136.5) | 0.020[a] |

Abbreviations: BA = Biliary atresia, AST = Aspartate aminotransferase, ALT = Alanine transaminase, ALP = Alkaline phosphatase, γGT = γ-glutamyl transferase.

Values are presented as median (interquartile ranges).

P-values by Mann-Whitney U test ('Fisher's exact test).

[a] P-value < 0.05.

**Table 2. Comparison of MRI parameters between BA and non-BA groups.**

|  | BA (n = 12) | non-BA (n = 5) | p-value |
|---|---|---|---|
| Area of ROIs (mm$^2$) | 1215.5 (1023.5–1407.5) | 1256.2 (958.7–1553.7) | 0.810 |
| ADC 10 ($10^{-3}$ mm$^2$/s) | 0.985 (0.854–1.117) | 1.332 (1.128–1.535) | 0.008 [a] |
| ADC 2 ($10^{-3}$ mm$^2$/s) | 0.987 (0.837–1.136) | 1.335 (1.104–1.567) | 0.017 [a] |
| D ($10^{-3}$ mm$^2$/s) | 0.835 (0.703–0.966) | 1.025 (0.821–1.228) | 0.115 |
| D* ($10^{-3}$ mm$^2$/s) | 33.2 (23.1–43.3) | 55.3 (39.7–71.0) | 0.021 [a] |
| f (%) | 13.4 (10.1–16.6) | 22.1 (17.1–27.2) | 0.007 [a] |
| DDC ($10^{-3}$ mm$^2$/s) | 0.889 (0.721–1.057) | 1.323 (1.062–1.584) | 0.009 [a] |
| α | 0.699 (0.634–0.765) | 0.556 (0.454–0.658) | 0.023 [a] |

Abbreviations: BA = Biliary atresia, ROI = Region-of-interest, ADC = Apparent diffusion coefficient, D = Slow true diffusion from pure water molecular diffusion, D* = Fast pseudo-diffusion from microcirculation and perfusion, f = Perfusion fraction, DDC = Distributed diffusion coefficient, α = Heterogeneity index.

Values are presented as estimated average values with 95% confidence intervals based on a linear mixed model.

[a] P-value < 0.05.

0.985 ×$10^{-3}$ mm$^2$/s vs. 1.332 ×$10^{-3}$ mm$^2$/s, p = 0.008) and the ADC 2 (median 0.987 ×$10^{-3}$ mm$^2$/s vs. 1.335 ×$10^{-3}$ mm$^2$/s, p = 0.017) were significantly lower in the BA group compared to the non-BA group. From BEM, D* (median 33.2 ×$10^{-3}$ mm$^2$/s vs. 55.3 ×$10^{-3}$ mm$^2$/s, p = 0.021) and *f* values (median 13.4%, vs. 22.1%, p = 0.009) were significantly lower in the BA group. However, the D value, which represented true water molecular diffusion, was not significantly different between the two groups (0.835 ×$10^{-3}$ mm$^2$/s vs. 1.025 ×$10^{-3}$ mm$^2$/s, p = 0.115). From SEM, DDC was significantly lower in the BA group (median 0.889 ×$10^{-3}$ mm$^2$/s vs. 1.323 ×$10^{-3}$ mm$^2$/s, p = 0.009), and the α value was significantly higher in the BA group (median 0.699 vs. 0.556, p = 0.023).

## Assessment and comparison of diagnostic performances

The diagnostic performances of significant results in laboratory and diffusion parameters are summarized in Table 3. The γGT value over 188 IU/L showed an AUC value of 0.867 (95% CI 0.616–0.984) for diagnosing BA. The ADC 10 value of ≤ 1.158 ×$10^{-3}$ mm$^2$/s showed an AUC value of 0.963, while The ADC 2 value of ≤ 1.165 ×$10^{-3}$ mm$^2$/s showed an AUC value of 0.925. In BEM, a D* value of ≤ 28.6 ×$10^{-3}$ mm$^2$/s showed an AUC value of 0.771, and an *f* value of ≤ 14.3% showed an AUC value of 0.850. In SEM, a DDC value of ≤ 1.256 ×$10^{-3}$ mm$^2$/s showed an AUC value of 0.925, while an α value over 0.680 had an AUC value of 0.788.

**Table 3. Diagnostic performance of parameters for the differentiation of BA from non-BA.**

|  | Cutoff values | Sensitivity (%) | Specificity (%) | AUC (95% CI) |
|---|---|---|---|---|
| γGT (IU/L) | > 188 | 75 (42.8–94.5) | 100 (47.8–100) | 0.867 (0.616–0.984) |
| ADC 10 ($10^{-3}$ mm$^2$/s) | ≤ 1.158 | 83.3 (62.6–95.3) | 100 (69.2–100) | 0.963 (0.834–0.998) |
| ADC 2 ($10^{-3}$ mm$^2$/s) | ≤ 1.165 | 79.2 (57.8–92.9) | 100 (69.2–100) | 0.925 (0.781–0.987) |
| D* ($10^{-3}$ mm$^2$/s) | ≤ 28.6 | 58.3 (36.6–77.9) | 90 (55.5–99.7) | 0.771 (0.595–0.897) |
| f (%) | ≤ 14.3 | 66.7 (44.7–84.4) | 100 (69.2–100) | 0.850 (0.686–0.949) |
| DDC ($10^{-3}$ mm$^2$/s) | ≤ 1.256 | 95.8 (78.9–99.9) | 80 (44.4–97.5) | 0.925 (0.781–0.987) |
| α | > 0.68 | 62.5 (40.6–81.2) | 90.0 (55.5–99.7) | 0.788 (0.614–0.908) |

Abbreviations: BA = Biliary atresia, AUC = Area-under-the-curve, CI = confidence interval, γGT = γ-glutamyl transferase, ADC = Apparent diffusion coefficient, D* = Fast pseudo-diffusion from microcirculation and perfusion, f = Perfusion fraction, DDC = Distributed diffusion coefficient, α = Heterogeneity index.

**Table 4. P-values comparing ROC curves for the differentiation of BA from non-BA.**

|  | ADC 10 | ADC 2 | D* | f | DDC | α |
|---|---|---|---|---|---|---|
| γGT | 0.125 | 0.380 | 0.340 | 0.847 | 0.389 | 0.396 |
| ADC 10 | . | 0.195 | 0.025 [a] | 0.073 | 0.181 | 0.033 [a] |
| ADC 2 | . | . | 0.083 | 0.286 | 1.000 | 0.130 |
| D* | . | | . | 0.416 | 0.075 | 0.859 |
| f | . | . | | . | 0.280 | 0.378 |
| DDC | . | . | . | | . | 0.125 |

Abbreviations: BA = Biliary atresia, ROC = Receiver operating characteristic, γGT = γ-glutamyl transferase, ADC = Apparent diffusion coefficient, D* = Fast pseudo-diffusion from microcirculation and perfusion, f = Perfusion fraction, DDC = Distributed diffusion coefficient, α = Heterogeneity index.

[a] P-value < 0.05.

When comparing ROC curves, ADC 10 showed higher diagnostic performances compared with D* and α (p = 0.025 and 0.033, respectively), while all other parameters showed no significant differences for diagnosing BA (Table 4). The diagnostic performance of ADC 2 using only two b-values showed excellent diagnostic performance and was not significantly different from that of γGT, ADC 10, f, and DDC for diagnosing BA. When we obtain the diagnostic performance of combining ADC 2 and γGT together, the AUC value was 0.987 with a 95% CI of 0.960–1.000.

## Discussion

In this study, only the γGT value (>188 IU/L) was a significant clinical and laboratory finding that differentiated BA from non-BA groups. However, most of the diffusion parameters including ADC 10, ADC 2, D*, f, and DDC values were significantly lower and the α value was significantly higher in the livers of those in the BA group compared to those in the non-BA group. Only the D value showed no significant difference suggesting that pure water molecular diffusion and vascular perfusion affected liver diffusion differently in the BA and non-BA groups. The diagnostic performances of most of the significant parameters were good to excellent, with the exception of the D* and α values. In addition, diagnostic performance of ADC 2 with b-values of 0 and 800 s/mm$^2$ was not significantly different from that of other parameters including ADC 10, $f$, DDC, and γGT; this shows that an ADC value with two b-values can be used to differentiate BA and non-BA while reducing image acquisition time in diffusion MRI of pediatric livers.

There are very few studies that have investigated DWI applied to liver MRIs for BA patients. In 2011 and 2015, two studies demonstrated that hepatic ADC values were significantly lower in children with BA compared to those with normal livers and that there was a negative correlation with the degree of liver fibrosis in BA patients [11,12]. Peng et al. mentioned that ADC values could also be used to predict the degree of liver fibrosis in postoperative patients [12]. In 2016, Liu et al. first utilized diffusion tensor imaging (DTI) to differentiate BA from non-BA and demonstrated that fractional anisotropy from DTI showed no significant differences between the two groups, while ADC values with 0 and 1000 s/mm$^2$ had significantly lower values in BA patients [13]. The diagnostic performance of ADC value was demonstrated by an AUC value of 0.805 with a cutoff value of 1.317 ×10$^{-3}$ mm$^2$/s with a sensitivity of 75% and specificity of 82% [13]. To our knowledge, our study is the first attempt to utilize MEM, BEM, and SEM DWI to differentiate BA from non-BA. Our results are consistent with other studies that showed a lower ADC value in the BA group; however we used a control group with non-

BA patients instead of normal children as was done in other studies. In addition, our diagnostic performances using these three models were higher than those in previous study.

BEM studies in adults with liver cirrhosis showed that either $D^*$ or $f$ was significantly lower in cirrhotic livers than in healthy livers, while D showed frequent trends for no significant difference or poor correlation with fibrosis level [18,19]. One recent study on pediatric liver BEM, $D^*$, and f values were significantly decreased in liver fibrosis [20]. We can suggest that decreased values in perfusion related parameters of BA patients could be from portal hypertension and decreased portal perfusion from increased collagen fibers and activated stellate cells in fibrotic livers [19]. Decreased microperfusion parameters from BEM were also noted in BA patients after receiving Kasai operation [20]. However, BA was known to have hepatic arteriopathy and increased hepatic arterial flow on color Doppler US could be used for the diagnosis of BA [21]. Effects of decreased portal perfusion, hepatic arteriopathy, and cholestasis in the BA patients before receiving operation were not fully understood yet. Therefore, further studies to know the reason for decreased perfusion parameters in BA patients are needed with pathologic correlation.

One recent study utilized SEM to assess liver fibrosis in adults [15]. They demonstrated that $D^*$, $f$, DDC, and α values were significantly decreased as the degree of liver fibrosis increased, while ADC and D did not show any differences [15]. Our results correspond with this study showing significantly lower $D^*$, $f$ and DDC values in the BA group. One of the interesting points in our study was that the α value was significantly higher in the BA group compared to the non-BA group. This might be because of differences between our study's samples compared to the other study. We could suggest that increased α values in BA group might be from relative homogeneity in decreased hepatic diffusion in the BA group compared to heterogeneous diffusion tendencies in the non-BA group, while higher α value represents decreased heterogeneity of diffusion rates. Our study showed the clinical potential of α value as a new quantitative MRI parameter for assessing diffusion heterogeneity. Because of the lack of studies applying SEM in pediatric liver and for BA patients, more studies are needed to validate this finding.

The $D^*$ showed relatively lower diagnostic performance, which was in accord with other studies that showed larger variation in $D^*$ measurement among BEM parameters [15,19,22]. The ADC 2 with two b-values showed excellent diagnostic performance and the diagnostic performance was not significantly different from that of other significant parameters in this study, suggesting that image acquisition time for obtaining MEM in pediatric liver can be reduced. In addition, this study demonstrated the potential utility of a new objective imaging parameter using MRI for the diagnosis of BA. Although γGT levels can be used to diagnose BA, imaging studies are usually recommended for the evaluation of infants with cholestasis due to overlaps in the laboratory results from other diseases [23]. In addition, MRI is more objective than US, which is a great advantage and research has emphasized the additive role of MRI for reducing the false-negative and false-positive results of US [4,5]. In addition, MRI can be used not only for morphologic evaluation, but to assess fibrosis in BA patients using DWI [12]. Usually the multimodality approach including laboratory and imaging studies is needed to accurately diagnose BA. Therefore, this study showed the potential utility of ADC and different DWI models as objective and quantitative imaging markers for the diagnosis of BA in conjunction with conventional laboratory results. Because this was a preliminary study with a small number of subjects, further investigation dealing with the additive role of different DWI models to conventional diagnostic methods is needed in the future.

Our study has several limitations. First, because it was retrospective, there was small number of included patients, especially in the non-BA group. Second, we did not evaluate the diagnostic performance of parameters according to the degree of liver fibrosis in BA patients or the

prognostic value of the parameters after the Kasai operation. We did not assess whether parameters had additive effects for diagnosing BA on conventional ultrasonography or MRI because we wanted to focus on demonstrating feasibility and utility of diffusion models in pediatric liver MRI and for BA patients. Further studies are needed that utilize these diffusion models to diagnose and predict prognostic values in BA patients. In addition, a multi-parametric approach that combines diffusion parameters with γGT, which showed equivalent diagnostic performance to diffusion parameters in our study, may also yield good performance and should be considered in future studies.

## Conclusions

The MEM, BEM, and SEM DWI were all feasible for pediatric liver MRI. Patients in the BA group had significantly lower ADC 10, ADC 2, $D^*$, $f$, and DDC values and higher γGT and α values in the liver than those in the non-BA group. The diagnostic performance of ADC 2 using only two b-values was excellent and was not significantly different from the diagnostic performances of γGT value and other diffusion parameters for diagnosing BA. Further prospective and larger studies are necessary for the clinical application of diffusion parameters in BA patients.

## Supporting information

**S1 File. Anonymized data of this study.**
(PDF)

## Author Contributions

**Conceptualization:** Jisoo Kim, Mi-Jung Lee, Myung-Joon Kim, Hyun Joo Shin.

**Data curation:** Jisoo Kim, Haesung Yoon, Mi-Jung Lee, Myung-Joon Kim, Kyunghwa Han, Seok Joo Han, Hong Koh, Seung Kim, Hyun Joo Shin.

**Formal analysis:** Jisoo Kim, Haesung Yoon, Kyunghwa Han, Hyun Joo Shin.

**Investigation:** Jisoo Kim, Mi-Jung Lee, Myung-Joon Kim, Seok Joo Han, Hyun Joo Shin.

**Methodology:** Jisoo Kim, Haesung Yoon, Mi-Jung Lee, Kyunghwa Han, Hyun Joo Shin.

**Project administration:** Hyun Joo Shin.

**Resources:** Myung-Joon Kim, Seok Joo Han, Hong Koh, Seung Kim, Hyun Joo Shin.

**Software:** Haesung Yoon, Myung-Joon Kim, Hyun Joo Shin.

**Supervision:** Mi-Jung Lee, Hyun Joo Shin.

**Validation:** Hyun Joo Shin.

**Visualization:** Hyun Joo Shin.

**Writing – original draft:** Jisoo Kim, Hyun Joo Shin.

**Writing – review & editing:** Jisoo Kim, Haesung Yoon, Mi-Jung Lee, Myung-Joon Kim, Kyunghwa Han, Hyun Joo Shin.

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
