## [Decision Letter · Decision Letter 0]

13 Sep 2019

PONE-D-19-21097

Usefulness of mono-, bi-, and stretched exponential model diffusion weighted imaging for the differentiation of biliary atresia and non-biliary atresia in pediatric liver MRI

PLOS ONE

Dear Dr Hyun Joo Shin 

Thank you for submitting your manuscript to PLOS ONE. After careful consideration, we feel that it has merit but does not fully meet PLOS ONE’s publication criteria as it currently stands. Therefore, we invite you to submit a revised version of the manuscript that addresses the points raised during the review process.

Please address the issues raised by the reviewers listed below.

We would appreciate receiving your revised manuscript by Oct 28 2019 11:59PM. To enhance the reproducibility of your results, we recommend that if applicable you deposit your laboratory protocols in protocols.io, where a protocol can be assigned its own identifier (DOI) such that it can be cited independently in the future. For instructions see: http://journals.plos.org/plosone/s/submission-guidelines#loc-laboratory-protocols

We look forward to receiving your revised manuscript.

Kind regards,

Gregory Tiao, M.D.

Academic Editor

PLOS ONE

Journal Requirements:

Additional Editor Comments (if provided):

The authors present an interesting manuscript but need to address the issues raised by the reviewers

Reviewers' comments:

Reviewer's Responses to Questions

**Comments to the Author**

1. Is the manuscript technically sound, and do the data support the conclusions?

Reviewer #1: Partly

Reviewer #2: Yes

2. Has the statistical analysis been performed appropriately and rigorously? 

Reviewer #1: Yes

Reviewer #2: Yes

3. Have the authors made all data underlying the findings in their manuscript fully available?

Reviewer #1: Yes

Reviewer #2: Yes

4. Is the manuscript presented in an intelligible fashion and written in standard English?

Reviewer #1: Yes

Reviewer #2: Yes

5. Review Comments to the Author

Reviewer #1: PONE-D-19-21097: Usefulness of mono-, bi-, and stretched exponential model diffusion weighted imaging for the differentiation of biliary atresia and non-biliary atresia in pediatric liver MRI

This manuscript demonstrates feasibility and utility of multi-b-value diffusion imaging in infant population for differentiation of biliary atresia and non-biliary atresia and provides further insights into the diffuse changes in liver. One important finding of the study as written in discussion is “The ADC 2 with two b-values showed excellent diagnostic performance and was comparable with other significant parameters in this study, suggesting that image acquisition time for obtaining MEM in pediatric liver can be reduced.” Performing MRI in infants is especially challenging due to time constraints imposed by the need for sedation/anesthesia. In that light this reviewer would recommend authors to keep the primary focus of the manuscript (and Title) something like “Clinical utility of two b-values compared to mono-, bi-, and stretched exponential model diffusion weighted imaging for the differentiation of biliary atresia and non-biliary atresia in infant liver MRI”.

The systematic comparison of using 2 (0 and 800) versus 10 (0, 25, 50, 75 100, 150, 200, 400, 600, 800) b values and using mono-, bi-, and stretched exponential models to derive different measures of diffusion and pseudo-diffusion is an important endeavor. As authors have described in the discussion prognostic value of the additive quantitative information from ADC mono, f, and DDC cannot be inferred from this specific study. Additionally, in routine clinical practice added value of ADC 2 (with considerations for sedation and expensive MRI) over γGT, itself may need justification.

Paper is very well written paper. Although this is a limited sample study, the findings of the study are very promising and encouraging for further more rigorous clinical validation. Authors have done a very good job of putting their contribution in this field in right perspective in the discussion session. Overall, discussion is very well written and has good balance. However, the points from the discussion do not reflect in the conclusion. The findings do not support claim of “new imaging parameters for the diagnosis of BA”. The γGT should be mentioned in the conclusion. Same applies for the Abstract of the manuscript.

Few remarks,

Title

Title may benefit with a qualifier

Clinical utility of two b-values compared to mono-, bi-, and stretched exponential model diffusion weighted imaging for the differentiation of biliary atresia and non-biliary atresia in infant liver MRI

Abstract

The values and differentiation power of should be reported. See comments above.

Introduction

A line referring to consideration for added time for extra sequences whould add to the context of the work (ADC 2 v/s ADC mono)

Materials and Methods

L 86: Consider changing different liver MRI DWI sequences to “composition of b values”

L 129: ADC “2” refers to ADC using 2 b values, thus ADC 1 may be better referred to as ADC mono or something like that.

Discussion

Overall, discussion is very well written and has good balance.

Figures.

Please include appropriate values and units for all the color bar

Reviewer #2: PONE-D-19-21097

• Very small study – should be considered a pilot investigation

• Study is innovative, with novel use of quantitative MRI parameters for detecting BA

• Results are compelling and deserving of a larger study

• Scattered corrections, additions are suggested to improve this manuscript

Abstract:

1. Define all abbreviations, please

2. Methods – what statistical test was used to compare groups – student’s t-test or Mann-Whitney U?

3. Re: Results – 1st sentence, consider providing actual results/p-values

4. Conclusion… can help in diagnosis of BA, perhaps… will certainly be some false positives and negatives

5. Keywords: change child to infant?; use “magnetic resonance imaging” and “diffusion-weighted imaging” as keywords

Introduction

6. MRI is not good for bile ducts in babies, in general… note, US actually has considerably better spatial resolution that MRI

7. When you discuss ultrasound, it may be worth separating gray-scale from shear wave elastography… both have been shown to add value in the diagnosis of BA

8. Any a priori hypothesis?

Methods

9. What is meant by different DWI sequences? Clearly list inclusion and exclusion criteria

10. Were labs really at exact time of MRI exam? Or did you use values closest to MRI?

11. Philips Healthcare MRI is out of Best, the Netherlands

12. Was DWI free breathing or respiratory-triggered/navigator gated?

13. What type of fat suppression was employed?

14. Who performed image analyses? Were they blinded to BA vs. non-BA diagnosis?

15. Why were linear mixed models needed to compare BA to non-BA pts?

Results

16. How was optimal ROC cut-off value chosen? Should you maximize sensitivity over specificity?

17. Does statistical analysis section indicate how ROC AUC values were compared?

18. Can you combine multiple MRI parameters and GGT to get even better performance (so called multi-parametric approach, use logistic regression which will generate an ROC AUC)

Discussion

19. OK

Conclusion

20. Emphasize that additional larger, prospective investigations are needed

References

21. OK

Figures

22. Fig 1F – can x-axis be labeled every 100? Is y-axis signal attenuation or signal intensity?

6. PLOS authors have the option to publish the peer review history of their article (what does this mean?). If published, this will include your full peer review and any attached files.

Reviewer #1: No

Reviewer #2: No

---

## [Author Response · Author response to Decision Letter 0]

20 Oct 2019

Responses to reviewers

Dear Editor,

Thank you for your considerate review and suggestions for the revision of our manuscript entitled “Usefulness of mono-, bi-, and stretched exponential model diffusion weighted imaging for the differentiation of biliary atresia and non-biliary atresia in pediatric liver MRI”. We have reviewed the suggestions made by the Reviewers and have done our best to revise the manuscript accordingly. Please find our responses below.

Reviewer’s Comments to Author:

Reviewer #1:

1. This manuscript demonstrates feasibility and utility of multi-b-value diffusion imaging in infant population for differentiation of biliary atresia and non-biliary atresia and provides further insights into the diffuse changes in liver. One important finding of the study as written in discussion is “The ADC 2 with two b-values showed excellent diagnostic performance and was comparable with other significant parameters in this study, suggesting that image acquisition time for obtaining MEM in pediatric liver can be reduced.” Performing MRI in infants is especially challenging due to time constraints imposed by the need for sedation/anesthesia. In that light this reviewer would recommend authors to keep the primary focus of the manuscript (and Title) something like “Clinical utility of two b-values compared to mono-, bi-, and stretched exponential model diffusion weighted imaging for the differentiation of biliary atresia and non-biliary atresia in infant liver MRI”.

Thank you for your detailed comments. We agree with the Reviewer’s recommendations and have revised the title, purpose, and conclusion to reflect the Reviewer’s comments.

2. The systematic comparison of using 2 (0 and 800) versus 10 (0, 25, 50, 75 100, 150, 200, 400, 600, 800) b values and using mono-, bi-, and stretched exponential models to derive different measures of diffusion and pseudo-diffusion is an important endeavor. As authors have described in the discussion prognostic value of the additive quantitative information from ADC mono, f, and DDC cannot be inferred from this specific study. Additionally, in routine clinical practice added value of ADC 2 (with considerations for sedation and expensive MRI) over γGT, itself may need justification.

When diagnosing biliary atresia, laboratory results including γGT can be used, but imaging modalities such as ultrasonography and MRI also have important roles because laboratory results of BA overlap with those of other diseases [1]. MRI is more objective compared to ultrasonography and research has emphasized the additive role of MRI for reducing the false-negative and false-positive results of ultrasonography [2,3]. In addition, not only for morphologic imaging, MRI can be used to assess the degree of fibrosis in biliary atresia patients using DWI [4]. This study showed that ADC and different DWI models have the potential to be objective and quantitative imaging markers for the diagnosis of BA in conjunction with conventional laboratory results. Because this study is a preliminary study with a small number of subjects, further investigation dealing with the additive role of different DWI models to conventional diagnostic methods is needed. We added a related explanation to the Discussion in this revision with the relevant references. 

3. Paper is very well written paper. Although this is a limited sample study, the findings of the study are very promising and encouraging for further more rigorous clinical validation. Authors have done a very good job of putting their contribution in this field in right perspective in the discussion session. Overall, discussion is very well written and has good balance. However, the points from the discussion do not reflect in the conclusion. The findings do not support claim of “new imaging parameters for the diagnosis of BA”. The γGT should be mentioned in the conclusion. Same applies for the Abstract of the manuscript.

Thank you for your encouraging comments and we have revised the Abstract, Discussion, and our conclusions to better reflect the findings of our study. 

Title

4. Title may benefit with a qualifier

Clinical utility of two b-values compared to mono-, bi-, and stretched exponential model diffusion weighted imaging for the differentiation of biliary atresia and non-biliary atresia in infant liver MRI

We agree with the Reviewer that the title can be more narrowed down and revised it accordingly.

Abstract

5. The values and differentiation power of should be reported. See comments above.

The requested data were added to the Abstract.

Introduction

6. A line referring to consideration for added time for extra sequences whould add to the context of the work (ADC 2 v/s ADC mono)

Thank you for your insightful comment. We added a brief mention of this advantage to the Introduction.

Materials and Methods

7. L 86: Consider changing different liver MRI DWI sequences to “composition of b values”

We revised not only the phrase in question but the overall sentence, because the prior expression was thought to be too ambiguous. 

8. L 129: ADC “2” refers to ADC using 2 b values, thus ADC 1 may be better referred to as ADC mono or something like that.

Thank you for pointing this out as we realized that the terms themselves were causing confusion. The mono-exponential model was used to obtain ADC 1 and ADC 2 with different b-values. To avoid confusion on these two ADC values, we changed ‘ADC 1’ to ‘ADC 10’ throughout the manuscript to indicate that the ADC value was obtained from 10 b-values. 

Discussion

9. Overall, discussion is very well written and has good balance.

Figures.

10. Please include appropriate values and units for all the color bar

Thank you for your comment. The units in each map were depicted by the software and we could not change the way the color bars or the units were displayed on the maps. However, in the manuscript, we presented each parameter value in widely accepted units. Therefore, we added each unit around the color bar for better clarity. In addition, the annotation in Figure 1F was revised (ADC 1 was changed to ADC 10). 

Reviewer #2: 

• Very small study – should be considered a pilot investigation

• Study is innovative, with novel use of quantitative MRI parameters for detecting BA

• Results are compelling and deserving of a larger study

• Scattered corrections, additions are suggested to improve this manuscript

Thank you for your supportive comments and hope our revisions will be found satisfactory.

Abstract:

1. Define all abbreviations, please

We wrote out all abbreviations in the Abstract as requested.

2. Methods – what statistical test was used to compare groups – student’s t-test or Mann-Whitney U?

The Mann-Whitney U test and Fisher’s exact test were used. We added these details to the Abstract.

3. Re: Results – 1st sentence, consider providing actual results/p-values

We added the actual results and p-values as requested.

4. Conclusion… can help in diagnosis of BA, perhaps… will certainly be some false positives and negatives

In light of Reviewer 1’s comments, we changed our conclusion to emphasize the diagnostic performance of ADC 2 with two b-values. DWI parameters could also lead to false-positive or false-negative results, but this preliminary study could not cover all of these parameters. Of its many traits, MRI is beneficial or advantageous because of its objectiveness and because it results in fewer false-positive or false-negative cases compared to ultrasonography [2]. Further investigation with a larger number of subjects is needed to validate the DWI models and conventional imaging modalities following this study. We added this content to the Abstract and the Discussion as well. 

5. Keywords: change child to infant?; use “magnetic resonance imaging” and “diffusion-weighted imaging” as keywords

We changed the Keywords as suggested.

Introduction

6. MRI is not good for bile ducts in babies, in general… note, US actually has considerably better spatial resolution that MRI

We truly appreciate your comment. We agree that US has good spatial resolution in infants. However, a previous study showed that MRI is not inferior to US when assessing findings such as triangular cord thickness, visibility of common bile duct and abnormality of gallbladder, which are used to diagnose biliary atresia [3]. In addition, MRI has strengths such as operator independency, reproducibility, and unrestricted FOV, unlike US [2]. We added and revised our explanation in the Introduction to reflect these previous studies. 

7. When you discuss ultrasound, it may be worth separating gray-scale from shear wave elastography… both have been shown to add value in the diagnosis of BA

We totally agree with the Reviewer and have changed the sentence accordingly.

8. Any a priori hypothesis?

There has been several papers on the utility of ADC using 2 b-values for the diagnosis of BA, while there are currently no studies on the use of bi- or stretched exponential models for the diagnosis of BA. Instead, there have been a few studies using the bi- or stretched exponential models for diagnosis or grading of fibrosis in children and adults. Since we thought that the ADC values obtained by using 2 b-values decrease due to fibrotic change in BA patients, we tried to investigate the diagnosis of BA using the mono, bi- and stretched exponential models. Also, if several of the model parameters had significant results, we wanted to identify the most efficient parameter. We added further explanations to the Introduction. 

Methods

9. What is meant by different DWI sequences? Clearly list inclusion and exclusion criteria

DWI MRI that were obtained without using the 10 b values mentioned in the Methods (0, 25, 50, 75, 100, 150, 200, 400, 600, and 800 s/mm2) were excluded. We revised the sentence because the prior expression was thought to be too ambiguous in light of the Reviewer’s comments.

10. Were labs really at exact time of MRI exam? Or did you use values closest to MRI?

Lab tests were performed within 3 days of the MRI examination. We added this detail to the Materials and Methods.

11. Philips Healthcare MRI is out of Best, the Netherlands

Thank you for pointing out this error. We immediately edited the MRI information.

12. Was DWI free breathing or respiratory-triggered/navigator gated?

Free breathing was the chosen method and we added this detail to the Materials and Methods.

13. What type of fat suppression was employed?

Gradient reversal fat suppression was employed and this detail was also added to the Materials and Methods.

14. Who performed image analyses? Were they blinded to BA vs. non-BA diagnosis?

One experienced pediatric radiologist who was blinded to the final diagnosis performed the image analyses. We added this information to the Materials and Methods.

15. Why were linear mixed models needed to compare BA to non-BA pts?

As ROIs were drawn on each of the two representative axial images of the liver separately, two values were obtained for each parameter. We used the linear mixed model to use each of the repeated measurements, instead of using median or mean values, for a more accurate analysis. We added further explanations on this to the Methods. 

Results

16. How was optimal ROC cut-off value chosen? Should you maximize sensitivity over specificity?

The optimal cutoff value was chosen to maximize q the sum of sensitivity and specificity. We added explanations to the Materials and Methods. 

17. Does statistical analysis section indicate how ROC AUC values were compared?

For the comparison of ROC curves, the DeLong method was used with the MedCalc program. We added this detail to the Statistical Analysis section.

18. Can you combine multiple MRI parameters and GGT to get even better performance (so called multi-parametric approach, use logistic regression which will generate an ROC AUC)

Thank you for your perceptive comment. Using logistic regression, we obtained the AUC value of 0.987 (95% CI 0.960-1.000) for combined ADC 2 and γGT. We presented this result in the Materials and Methods and Results. As we could not compare the diagnostic performances of multiparametric results combining MRI parameters, laboratory tests, and even conventional imaging findings in this study, further research following this preliminary study is needed for validation. We added a statement on the necessity for such future research to the Discussion as well. 

Discussion

19. OK

Conclusion

20. Emphasize that additional larger, prospective investigations are needed

Because our study is of retrospective design and small in size, we agree that larger prospective studies are necessary before diffusion parameters can be clinically applied in pediatric liver MRI. We added these details to the Conclusion.

References

21. OK

Figures

22. Fig 1F – can x-axis be labeled every 100? Is y-axis signal attenuation or signal intensity?

The software for analyzing each parametric map automatically generated the graph (Fig 1F) and we could not change the scales of the X-axis. The X-axis represents b-values (0, 25, 50, 75, 100, 150, 200, 400, 600, and 800 s/mm2) and the Y-axis represents diffusion-related signal attenuation. A higher number of b-values makes more pronounced diffusion-related signal attenuation. We added this information to the Figure legend. 

Thank you once again for your time and efforts in reviewing our manuscript.

References

1. Sun S, Chen G, Zheng S, Xiao X, Xu M, Yu H, et al. Analysis of clinical parameters that contribute to the misdiagnosis of biliary atresia. J Pediatr Surg. 2013;48: 1490-1494. doi:10.1016/j.jpedsurg.2013.02.034 PMID:23895960

2. Kim YH, Kim MJ, Shin HJ, Yoon H, Han SJ, Koh H, et al. MRI-based decision tree model for diagnosis of biliary atresia. Eur Radiol. 2018. doi:10.1007/s00330-018-5327-0 PMID:29476221

3. Han SJ, Kim MJ, Han A, Chung KS, Yoon CS, Kim D, et al. Magnetic resonance cholangiography for the diagnosis of biliary atresia. J Pediatr Surg. 2002;37: 599-604 PMID:11912518

4. Peng SS, Jeng YM, Hsu WM, Yang JC, Ho MC. Hepatic ADC map as an adjunct to conventional abdominal MRI to evaluate hepatic fibrotic and clinical cirrhotic severity in biliary atresia patients. Eur Radiol. 2015;25: 2992-3002. doi:10.1007/s00330-015-3716-1 PMID:25921590

---

## [Decision Letter · Decision Letter 1]

4 Dec 2019

Clinical utility of mono-exponential model diffusion weighted imaging using two b-values compared to the bi- or stretched exponential model for the diagnosis of biliary atresia in infant liver MRI

PONE-D-19-21097R1

Dear Dr. Shin,

We are pleased to inform you that your manuscript has been judged scientifically suitable for publication and will be formally accepted for publication once it complies with all outstanding technical requirements.

With kind regards,

Gregory Tiao, M.D.

Academic Editor

PLOS ONE

Additional Editor Comments (optional):

The authors have addressed all the concerns raised during the review

Reviewers' comments:

Reviewer's Responses to Questions

**Comments to the Author**

1. If the authors have adequately addressed your comments raised in a previous round of review and you feel that this manuscript is now acceptable for publication, you may indicate that here to bypass the “Comments to the Author” section, enter your conflict of interest statement in the “Confidential to Editor” section, and submit your "Accept" recommendation.

Reviewer #1: All comments have been addressed

Reviewer #2: All comments have been addressed

2. Is the manuscript technically sound, and do the data support the conclusions?

Reviewer #1: Yes

Reviewer #2: Yes

3. Has the statistical analysis been performed appropriately and rigorously? 

Reviewer #1: Yes

Reviewer #2: Yes

4. Have the authors made all data underlying the findings in their manuscript fully available?

Reviewer #1: Yes

Reviewer #2: Yes

5. Is the manuscript presented in an intelligible fashion and written in standard English?

Reviewer #1: Yes

Reviewer #2: Yes

6. Review Comments to the Author

Reviewer #1: (No Response)

Reviewer #2: Thank you for addressing the reviewers' comments. I have no remaining concerns. Your study, while small, has interesting results/conclusions and should pave the way for future investigations.

7. PLOS authors have the option to publish the peer review history of their article (what does this mean?). If published, this will include your full peer review and any attached files.

Reviewer #1: No

Reviewer #2: No

---

## [Editor Report · Acceptance letter]

10 Dec 2019

PONE-D-19-21097R1 

Clinical utility of mono-exponential model diffusion weighted imaging using two b-values compared to the bi- or stretched exponential model for the diagnosis of biliary atresia in infant liver MRI 

Dear Dr. Shin:

I am pleased to inform you that your manuscript has been deemed suitable for publication in PLOS ONE. Congratulations! Your manuscript is now with our production department. 

With kind regards,

on behalf of

Dr. Gregory Tiao 

Academic Editor

PLOS ONE